

# Predicting the satisfiability of Boolean formulas by incorporating gated recurrent unit (GRU) in the Transformer framework

Wenjing Chang, Mengyu Guo and Junwei Luo

School of Software, Henan Polytechnic University, Jiaozuo, Henan, China

## ABSTRACT

The Boolean satisfiability (SAT) problem exhibits different structural features in various domains. Neural network models can be used as more generalized algorithms that can be learned to solve specific problems based on different domain data than traditional rule-based approaches. How to accurately identify these structural features is crucial for neural networks to solve the SAT problem. Currently, learning-based SAT solvers, whether they are end-to-end models or enhancements to traditional heuristic algorithms, have achieved significant progress. In this article, we propose TG-SAT, an end-to-end framework based on Transformer and gated recurrent neural network (GRU) for predicting the satisfiability of SAT problems. TG-SAT can learn the structural features of SAT problems in a weakly supervised environment. To capture the structural information of the SAT problem, we encodes a SAT problem as an undirected graph and integrates GRU into the Transformer structure to update the node embeddings. By computing cross-attention scores between literals and clauses, a weighted representation of nodes is obtained. The model is eventually trained as a classifier to predict the satisfiability of the SAT problem. Experimental results demonstrate that TG-SAT achieves a 2%–5% improvement in accuracy on random 3-SAT problems compared to NeuroSAT. It also outperforms in SR(N), especially in handling more complex SAT problems, where our model achieves higher prediction accuracy.

# INTRODUCTION

The Boolean satisfiability (SAT) problem is a classic problem in mathematical logic, which involves determining whether there exists at least one assignment to the variables such that the given formula evaluates to true. If such an assignment exists, the SAT problem is satisfiable; otherwise, it is unsatisfiable. The SAT problem was proved to be NP-complete by *Cook (1971)*, and it is widely used in practical applications such as circuit design (*Kasi & Sarma, 2013*), scheduling (*Li et al., 2012*), combinatorial optimization (*Rintanen, 2012*), and planning (*Goldberg, Prasad & Brayton, 2001*). These real-world problems are generally

Corresponding author
Junwei Luo, luojunwei@hpu.edu.cn

transformed into SAT problems by different coding techniques and then the SAT solver is invoked to solve them (*Gomes et al., 2008*). Traditional solvers based on backtracking search have been able to solve problems of practical interest with millions of Boolean variables (*Silva, Lynce & Malik, 2021*). However, in the worst case, for some complex SAT instances, traditional solvers may face exponential time complexity. SAT solvers are usually limited by the size of the problem.

In recent years, machine learning techniques have emerged in the field of combinatorial optimization (*Bengio, 2021*), and many learning-based methods for solving SAT problems have been extensively studied. *Devlin & O'Sullivan (2008)* investigated many machine learning classification algorithms for predicting the satisfiability of SAT problems based on hand-designed features. However, hand-designed features are somewhat subjective and limited. To avoid this problem, *Hopfield & Tank (1985)* developed a well-known neural structure that solves a number of instances of the traveling salesman problem. Conventional system satisfiability logic suffers from the problem of inflexible logic structure leading to inadequate explanations. *Karim et al. (2021)* introduced the advantages of non-system satisfiability logic to improve the flexibility of the logical structure. Non-systematic satisfiability ignores the distribution of positive and negative literals in logical structures as neuronal representations, and *Zamri et al. (2022)* use genetic algorithms to assign unbiased distributions of negative literals. However, what the model actually learns is not transparent enough. *Someetheram et al. (2022)* used an Election Algorithm (EA) to find consistent interpretations that minimize the cost function of the proposed logic rules, which plays a vital role in creating interpretable artificial intelligence.

In addition, some work has used graph neural networks (GNN) to obtain embeddings of SAT problem features to predict their satisfiability. *Yolcu & Póczos (2019)* introduced GNN as a heuristic for variable selection in a stochastic local search algorithm. NLocalSAT improves the stochastic local search algorithm by changing the initial assignment through a neural network (*Zhang et al., 2020*). *Selsam & Bjørner (2019)* guided the CDCL solver by predicting the variables in the unsatisfied core. These works use neural networks as heuristic algorithms to guide traditional solvers, which improve solving efficiency but lack flexibility. *Bünz & Lamm (2017)* utilized graph neural networks to learn the structural features of SAT instances. NeuroSAT uses message passing networks to learn the satisfiability of random SAT instances (*Selsam et al., 2018*). DeepSAT applies mature knowledge in the field of electronic design automation to solve SAT problems (*Li et al., 2022*). *Liu et al. (2021)* were the first to use GNN to learn the solution of MaxSAT problems. They built two typical GNN models to study the ability of GNN learning to solve MaxSAT problems from both theoretical and practical perspectives. The results show that both models achieve higher accuracy on the test set and have satisfactory generalisation ability for larger and more difficult problems.

We focus on end-to-end methods to predict satisfiability. SAT instances that cannot be satisfied are generally due to variable conflicts in Boolean formulas, resulting in the inability to satisfy all constraint conditions. If a model wants to learn to identify modules that cause conflicts, the input needs to include information about the interrelationships

between variables. How to capture the structural information of SAT problems in the feature space is particularly important for studying end-to-end methods.

The Transformer model consists of a self-attention mechanism and a fully-connected feedforward neural network layer that efficiently captures the dependencies between input sequences. In natural language, attentional mechanisms can learn the importance of different words in a sentence and determine which words are more critical for understanding the meaning of the sentence (*Vaswani et al., 2017*). We investigate whether natural language processing models can be used for SAT problems by considering literals in SAT instances as words and clauses as sentences. the order of words affects the expression of a sentence. Unlike natural language, SAT instances that change the order of clauses or the literal order within clauses do not affect satisfiability. From this perspective, we may not be able to analogize Boolean formulas with natural language. However, we represent Boolean formulas in graphs and operate on nodes based on the topology of the graph, without specifying a specific node order. This provides hope for us to study natural language processing models suitable for SAT problems. In text classification tasks, a few key words can determine the theme of the text. Similar to natural language processing tasks, in Boolean formulas, we can deduce the assignment of the remaining variables by assigning some of them. By identifying these key variables, we can predict the satisfiability of the SAT problem.

In this article, we present for the first time TG-SAT, an end-to-end Transformer-based framework for predicting the satisfiability of SAT problems. Transformer learns the degree of semantic associations between different positions in an input sequence and the importance of different positions. Whereas GRU is more effective in dealing with long sequences (*Cho et al., 2014*), neither of them is able to aggregate information from higher-order neighboring nodes. Therefore, we enable each node to obtain information about its related higher-order neighboring nodes by incorporating a message passing network (*Gilmer et al., 2017*), and then compute the attention scores between literal and clause nodes to obtain the weighted embedding of the nodes. Finally, we use a multilayer perceptron for prediction. Experimental results show that our model outperforms NeuroSAT in prediction accuracy on both random SAT problems and random 3-SAT problems. The contributions of our work are summarized below:

- We propose a new neural network framework, TG-SAT, which incorporates message passing for predicting SAT problems. To obtain rich information about node embeddings, we use the self-attention mechanism to transfer information about variables sharing the same clauses or clauses containing the same variables, and obtain information about the nodes' multistep neighboring nodes through the message-passing network. In addition, we apply the cross-attention mechanism to optimize clause-to-literal and literal-to-clause message exchanges.
- We make the first attempt to incorporate GRUs into the Transformer architecture to predict stochastic SAT problems. We encode random SAT instances as graph structures, taking into account the importance of multi-order neighbor information between nodes, while traditional Transformer models usually do not have direct access to multi-order

neighbor node information. Therefore, we introduce the GRU module, which performs message aggregation on the neighboring clause nodes (literal nodes) of literal nodes (clause nodes), and obtains node embeddings containing the information of multiorder neighboring nodes after several iterations.

- We also explain the role of self-attention mechanisms and cross-attention mechanisms in predicting the satisfiability of SAT problems.

The rest of this article is organized as follows. In 'Related Work', we provide a brief review of the work. In 'Methods', we first outline the model designed, then an overview of the SAT problems and their graphical representations, and then describe the implementation details for each network layer of the model. In 'Experimental setup', the experimental data set and the associated settings for the experiment are described in detail, and the results are presented and analyzed in Results. 'Conclusions' and further research are presented in Conclusion.

## RELATED WORK

### Learning-based SAT solver

In recent years, many researchers have modeled the relationship between the characteristics of SAT problems and their satisfiability and used machine learning models to make predictions. *Guo et al. (2022)* summarized these approaches. *Devlin & O'Sullivan (2008)* treat an instance of a SAT problem as a vector with several features, each representing some attribute or property of the SAT problem. They transformed the solution of the SAT problem into a binary classification task on the feature vectors (*Devlin & O'Sullivan, 2008*). SATzilla uses a set of features to describe the properties of a SAT problem instance and uses a machine learning model to predict the most suitable solution algorithm for the features of the current problem instance (*Xu et al., 2008*). *Danisovszky, Yang & Kusper (2020)* constructed a 48-vectored elicitation set and, by selecting suitable features, the improved the classification performance of the machine learning model. Over the past decade or so, this method of extracting features from input formulas has achieved high accuracy on different benchmarks, but some of the structural features of SAT instances are lost in the feature extraction part. Deep neural networks are able to learn high-level feature representations of the data through multiple layers of nonlinear transformations, facilitating the prediction of satisfiability in an end-to-end framework.

*Bünz & Lamm (2017)* define a graph representation of the Conjunctive Normal Form (CNF) for Boolean formulas to explore the applicability of neural networks in the study of Boolean semantics, and the results show that in the absence of problem-specific feature engineering, a GNN can learn the structural features of SAT instances. Since then, many methods for solving SAT problems based on GNN have appeared. Boolean formulas have permutation invariants and negation invariants, and NeuroSAT encodes CNF formulas into literal clause graphs in order to learn these features, and can solve larger and more complex problems than in training by simply performing more messaging iterations (*Selsam et al., 2018*). Inspired by these efforts, we also studied graph representations of SAT

problems to explore whether other neural networks could also learn the features of SAT problems, such as deep learning models for natural language processing.

*Shi et al. (2021a)* first attempted to solve the MaxSAT problem using Transformer, which applied a transformer to aggregate messages on nodes. SATformer uses GNN to obtain clause embeddings in CNF. The hierarchical Transformer architecture is applied on the clause embedding to capture the relationship between clauses, give higher weight to clauses in the minimum unsatisfiable cores (MUC), and effectively learn the correlation between clauses (*Shi et al., 2022*). Based on Transformer, Heterogeneous Graph Transformer (HGT) defines the heterogeneous attention mechanism based on meta-paths for the self-attention between literals, the cross-attention based on the bipartite graph links between literals and clauses (*Shi et al., 2021b*). The experimental results show that solving SAT problems using the Transformer framework is more competitive and generalized than simple message passing networks. Our research focuses on improving the end-to-end framework to predict satisfaction. Our proposed model represents a given SAT instance as an undirected graph. First, the Transformer model is applied to obtain node embeddings, after which, to obtain information about multi-order neighboring nodes, we introduce a message passing network and update the node embeddings using a GRU network. Next, we optimize the message passing of heterogeneous nodes using the cross-attention mechanism to learn the invariant structural features of key variables. Finally, the updated node embeddings are used to predict the satisfiability of the problem.

## Random SAT problem and random 3-SAT problem

Currently, there are two general forms of SAT problems. One is derived from industrial problems through transformation, with significant limitations in terms of both quantity and structure. Consequently, such SAT instances are not utilized in our study to evaluate our model. The other type is randomly generated SAT problems. Instances of these SAT problems are generated using random generation algorithms, allowing for control over the difficulty and characteristics of the generated instances, such as satisfiability probabilities and phase transition phenomena. They can encompass problems of various difficulties and sizes in large-scale experiments. Thus, randomly generated SAT instances are commonly employed to assess the performance and efficiency of SAT solvers. Random 3-SAT problems are random SAT problems in which the number of literals in each clause is fixed to three. For random 3-SAT problems, even state-of-the-art conventional solvers still struggle to solve problems with hundreds of variables. Therefore, we primarily chose to use random SAT problems and random 3-SAT problems to evaluate the performance of our model.

## METHODS

The implementation of TG-SAT is shown in Fig. 1. (1) First, we encode the SAT problem as an undirected graph G, represented by an adjacency matrix. (2) The initial vectors are created for each literal node $l_i \in G$ and each clause node $c_i \in G$. (3) The self-attention score for each node is calculated, capturing the dependencies between nodes and the degree of attention paid to different locations. (4) Message passing for each node, in each iteration, each clause node receives its own weighted information and information about neighboring

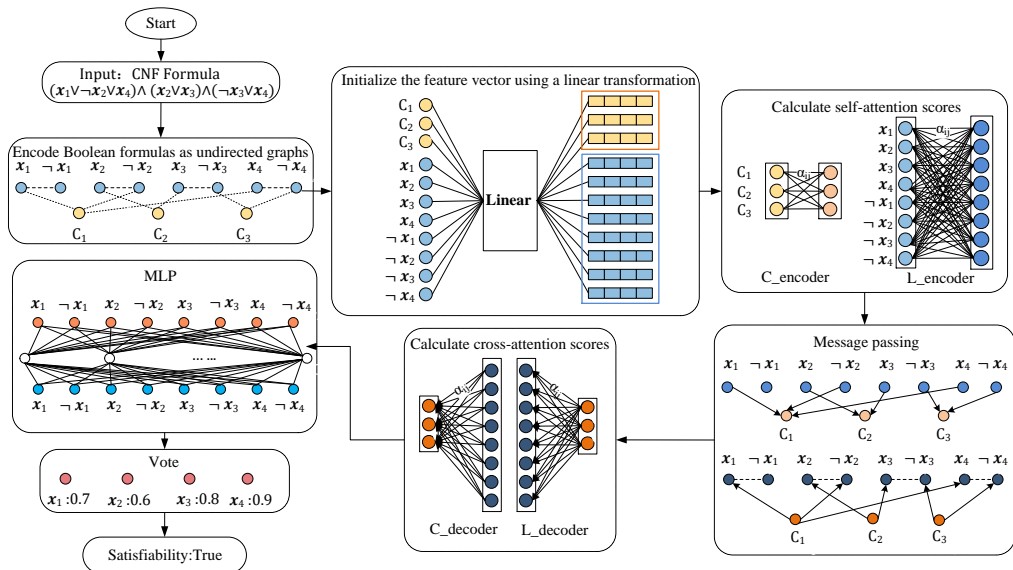

**Figure 1 The overview of our model, TG-SAT.**

literal nodes, and updates its embedding accordingly, and each literal node receives its own weighted information and information about neighboring clause nodes, and updates its embedding accordingly. (5) After $T$ iterations, TG-SAT learns the attentional weights $\alpha_{ij}$ between the literal nodes and clause nodes by cross-attention, The feature vectors of literal nodes and clause nodes are then weighted and averaged according to the attention weights, and obtain new feature vectors for literal nodes $l_i'$ and clause node $c_i'$. (6) Finally, the literal node $l_i'$ is entered into the MLP, and the vote for each literal is obtained by the activation function, The average of the votes is calculated and if it is greater than 0.5, it is considered satisfiable, otherwise it is considered unsatisfiable.

## Preliminaries
### *Boolean formula*
A Boolean formula consists of a set of variables $\{x_i\}_{j=1}^J \in \{True, False\}$ and logical operators $\{and \wedge, \ or \vee, \ no \neg\}$. $i$ represents the $i$th variable in a certain clause, and $j$ represents the $j$th clause. Since Boolean formulas can be transformed into equivalent conjunctive normal form(CNF) in linear time, we usually use the CNF to represent SAT problems (*Tseitin, 1983*). CNF specifies that variables and their negatives are called literals, with positive literals denoted $x_i$ and negative literals denoted $\neg x_i$. A disjunction of several literals constructs a clause, denoted $C_i = (x_1 \vee x_2 \vee \ldots)$. A conjunction of clauses forms the propositional instance, denoted $\phi = (C_1 \wedge C_2 \wedge \ldots)$. $P = (x_1 \vee \neg x_2 \vee x_4) \wedge (x_2 \vee x_3) \wedge (\neg x_3 \vee \neg x_4)$ as an example to explain CNF. $P$ contains 4 variables $I = \{x_1, x_2, x_3, x_4\}$ and 3 clauses $\{C_1 = (x_1 \vee \neg x_2 \vee x_4), C_2 = (x_2 \vee x_3), C_3 = (\neg x_3 \vee \neg x_4)\}$. In order for $P$ to be true, at least one literal in each clause takes the value true. If such a set of variable assignments exists, we call the problem satisfiable, called SAT, and otherwise unsatisfiable, denoted by UNSAT.

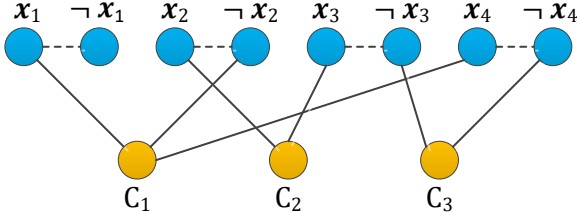

**Figure 2  Literal-clause graph of formulas.**

$$M = \begin{bmatrix} 1 & 0 & 0 & 1 & 0 & 1 & 0 & 0 \\ 0 & 1 & 1 & 0 & 0 & 0 & 0 & 0 \\ 0 & 0 & 0 & 0 & 0 & 0 & 1 & 1 \end{bmatrix}$$

**Figure 3  The adjacency matrix of P.**

### Formula graph

We know that variable conflicts in Boolean formulas can lead to unsatisfiability. To enable our model to learn to recognize these conflicts, it's essential to ensure that the model's input contains information about the relationships between the variables in the formula. Therefore, in this article, our model encodes CNF formulas as an undirected graph. In this graph representation, each literal is treated as a literal node, each clause as a clause node, and an edge is created if a literal occurs in a clause. Additionally, a different type of edge is created for each pair of complementary literals. This encoding strategy helps strengthen permutation invariance and negation invariance, aiding our model in better extracting structural features of SAT problems. The graphical representation of formula $P$ in 'Initialize node embedding' is shown in Fig. 2. The adjacency matrix for formula $P$ is shown in Fig. 3. $M_{ij} = 1$ indicates the presence of an edge between a literal node and a clause node, while $M_{ij} = 0$ indicates the absence of such an edge.

### Simulation design

This section provides a detailed description of the model construction process. Our network consists of three multilayer perceptrons ($L_{msg}, C_{msg}, L_{vote}$), two self-attention ($L_{enconder}, C_{en}coder$), two cross-attention ($L_{decoder}, C_{decoder}$), two GRUs ($L_{GRU}, C_{GRU}$) as shown in Fig. 4. Updating the node embedding consists of three phases.

(1) The multi-head self-attention mechanism captures the different positional and attention weight information of the clause nodes, and then each clause node obtains information from its neighboring literal nodes and updates the clause embedding accordingly. After T iterations, the clause nodes' embeddings will encompass both their own global information and information from multi-order neighbor nodes.

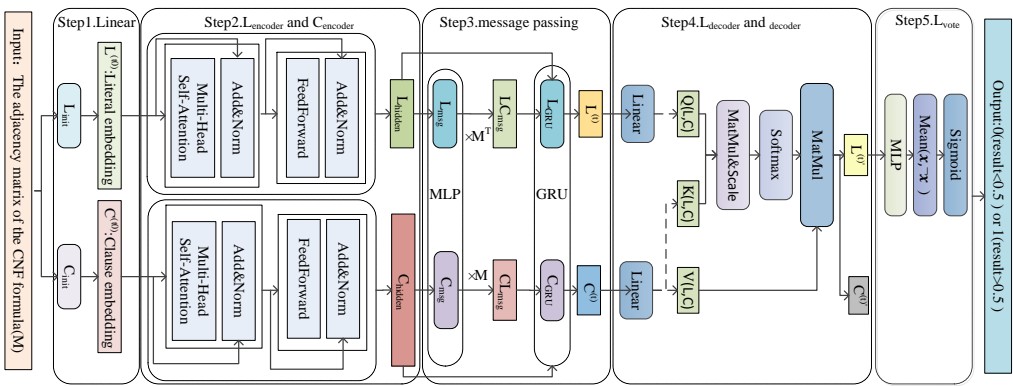

**Figure 4** Schematic diagram of the model.

(2) The multi-head self-attention mechanism captures the different positional information and attention weight information of the literal nodes, and then each literal node obtains the information from its neighboring clauses and complementary literals and updates the literal embedding accordingly. After T iterations, the literal nodes' embeddings will encompass both their own global information and information from multi-order neighbor nodes.

(3) The cross-attention mechanism is used to compute attention scores between the literal nodes and the clause nodes, resulting in weighted embeddings for all nodes. All nodes are updated in each iteration of TG-SAT

Our model operates on nodes and edges based on the graph's topology, and the order of nodes and edges does not influence the prediction results. This property aligns with the permutation invariance of SAT instances. The model's inputs consist of the adjacency matrices of an arbitrary number of literals and clauses, enabling training and evaluation on problems of varying sizes.

### Initialize node embedding

In SAT, only the number of variables and the logical relationships between them need to be considered. There is no need to consider the specific arrangement or values of the variables. Based on this property, we initialize each literal node $L_i$ as a vector $L_{init} \in R^d$, as shown in Eq. (1), and each clause node $C_j$ as another vector $C_{init} \in R^d$, as shown in Eq. (2), to obtain the initial embeddings of the literal and clause nodes, denoted as $L^{(t0)}$ and $C^{(t0)}$, respectively.

$$L^{(t0)} = L_i \cdot W_1^T + b_1. \tag{1}$$

$$C^{(t0)} = C_J \cdot W_2^T + b_2. \tag{2}$$

where $W_1$ and $W_2$ are trainable parameter matrices and $b_1$ and $b_2$ are bias vectors. $i$ represents the $i$th variable, and $j$ represents the $j$th clause.

### 3.2.2 Self-attention mechanism

To empower the node embeddings with ample expressive capability, the encoder layer ($L_{encoder}$ and $C_{encoder}$) computes self-attention for literal nodes and clause nodes, respectively, as outlined in Eqs. (3)–(6).

$$Q = XW_q, K = YW_k, V = YW_v. \tag{3}$$

$$\alpha_{i,j} = \frac{\langle q_i, k_j \rangle}{\sum_{n \in N_{(i)}} \langle q_i, k_n \rangle} = \text{SoftMax}\langle Q, \quad K^T \rangle. \tag{4}$$

$$v'_i = \sum_{j \in N_{(i)}} \alpha_{i,j} v_j. \tag{5}$$

$$\text{MultHeadAttention}(Q, K, V) = \text{concat}\left(\alpha^1_{i,j} v_1, \alpha^2_{i,j} v_2, \ldots \alpha^h_{i,j} v_h\right) W_3. \tag{6}$$

where $Q, K, V$ are query, key, value in transformer terminology. $W_q, W_k, W_v, W_3$ are trainable parameter matrices. $q_i$ is the $i$th row vector of the $Q$ matrix, $k_j$ is the $j$th column vector of the $K$ matrix, and $v_j$ is the $j$th row vector of the $V$ matrix. $\langle q, k \rangle = exp\left(\frac{qk^T}{\sqrt{d_1}}\right)$ is the inner product of two vectors. $d_1$ is the dimension of each header. $\alpha_{i,j}$ is the attention weight, which indicates the importance of the node. $v'_i$ is the weighted feature vector.

First, according to Eq. (3), we set $X = Y = Z = L^{(t0)}$ to obtain the $Q, K, V$ matrices for the literal nodes. Then, the self-attention weights between literal nodes are computed according to Eq. (4). Recognizing the challenge of capturing all node features using a single attention score, we employ a multi-head attention mechanism to capture global feature information of literal nodes. Specifically, attention weights computed for each head are multiplied by the corresponding value tensor, as described in Eq. (5). Subsequently, the results are concatenated and linearly mapped to obtain the final multi-head attention output, which represents the current embedding of the literal node, as shown in Eq. (6). Similarly, we set $X = Y = Z = C^{(t0)}$ to obtain the embedding of the clause nodes. Eventually, each node incorporates its own weighted information, and the updated embedding is used as input in the next message passing network layer. The specific implementation flow of the attention layer is illustrated in Fig. 5.

### Message passing mechanism

Self-attention mechanisms lack access to information about multi-order neighboring nodes. Since our goal is for the model to learn to identify conflicting modules that cause unsatisfiability, it's crucial to ensure that node embeddings contain rich information about the relationships between nodes. To address this challenge, we introduce a GRU-based message passing network. The specific calculation process of message passing is depicted in Eqs. (7) and (8).

$$C^{(t+1)}, C^{(t+1)}_h \leftarrow C_{\text{GRU}}\left(\left[C^{(t)}_h, M^T L_{msg}(L^{(t)})\right]\right). \tag{7}$$

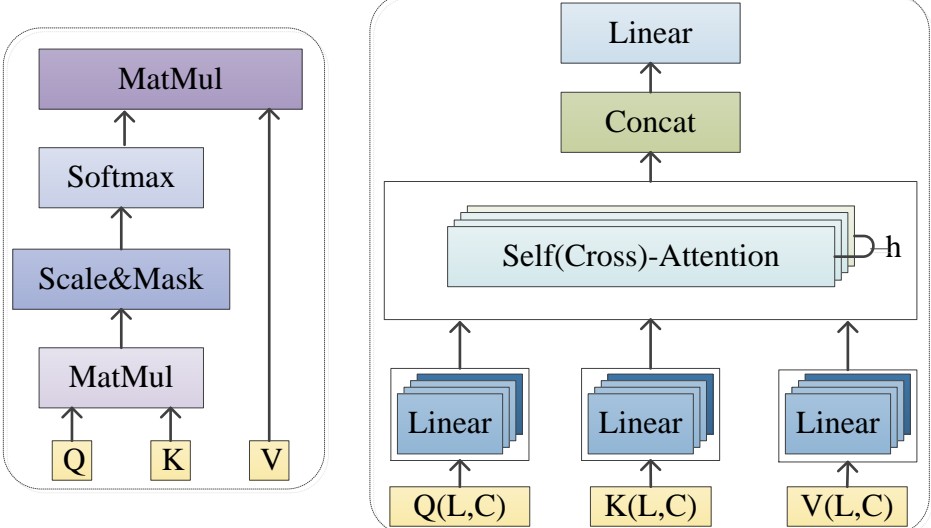

**Figure 5  Attention mechanisms and multiple attention mechanisms.**

$$L^{(t+1)}, L_h^{(t+1)} \leftarrow L_{GRU}\left(\left[L_h^{(t)}, Flip(L^{(t)}), MC_{msg}(C^{(t+1)})\right]\right). \tag{8}$$

Assuming the number of clauses is denoted by $m$, and the number of variables is denoted by $n$. The matrix $L^{(t)} \in \mathbb{R}^{(2n \times d)}$ represents the feature vectors of all literals, where each literal has a feature vector of dimension $d$. Similarly, $C^{(t)} \in \mathbb{R}^{(m \times d)}$ represents the feature vectors of all clauses. $L_h^{(t)} \in \mathbb{R}^{(2n \times d)}$ and $C_h^{(t)} \in \mathbb{R}^{(m \times d)}$ represent the hidden states of $L_{GRU}$ and $C_{GRU}$, respectively. The node embedding dimension is $d$, set to 128 in this article. $M(i,j) = 1 l_i \in c_j$ is the adjacency matrix, indicating whether literal $l_i$ is in clause $c_j$. Flip is a function that exchanges the vector representations of positive and negative literals.

The message passing process in each iteration consists of two phases. Firstly, to update the clause node embeddings, we utilize the clause node embeddings obtained from the $C_{encoder}$ layer as inputs to the hidden states of the GRU. These embeddings aggregate the information from neighboring literal nodes. The clause node embeddings are updated by the GRU. Secondly, to update the literal node embeddings, we use the literal node embeddings obtained from the $L_{encoder}$ layer as inputs to the hidden states of the GRU. Since the SAT problem exhibits negation invariance, our model ensures this property by swapping the positions of the embeddings of positive literal nodes with the embeddings of their corresponding negative literal nodes. The literal node embeddings, which aggregate information from neighboring clause nodes and their complementary literal nodes, are then inputted to the current state and updated *via* GRU. The GRU computation process involves computing the update gate $z$, the reset gate $r$, the candidate hidden state $h_t'$, and the final hidden state $h_t$, respectively, as shown in Eqs. (9)–(12).

$$z = \sigma\left(W_z \cdot \left[h_{t-1}, x_t\right] + b_z\right). \tag{9}$$

$$r = \sigma\left(W_r \cdot [h_{t-1}, x_t] + b_r\right). \tag{10}$$

$$h'_t = \tanh\left(W_h \cdot [r \odot h_{t-1}, x_t] + b_h\right). \tag{11}$$

$$h_t = (1-z) \odot h_{t-1} + z \odot h'_t. \tag{12}$$

where $W_z, W_r, W_h$ are the learnable weight matrices, $b_z, b_r, b_h$ are the bias vectors, $\sigma$ denotes the sigmoid activation function and $\odot$ denotes the element-by-element multiplication. $h_{t-1}$ is the hidden state of the previous time step and $x_t$ is the input of the current time step.

### Cross-attention mechanism

In each decoder $L_{decoder}$ and $C_{decoder}$, attention scores between literals and clauses are computed using a cross-attention mechanism, facilitating the exchange of weighted messages between different types of nodes. In $L_{decoder}$, cross-attention is computed by setting $X = L$ and $Y = C$, allowing literal nodes to receive weighted information from clause nodes. Conversely, in $C_{decoder}$, clause nodes receive weighted messages from literal nodes by setting $X = C$ and $Y = L$. The computational process mirrors self-attention as described in Eqs. (3), (4), (5). To enhance the representation of node feature vectors, we feed the attention-weighted node features into a feed-forward network. Ultimately, we obtain node embeddings $L^{(t)'}$ and $C^{(t)'}$ containing the weighted structural information.

### Vote mechanism

In summary, we know that after simple message passing, the embedding of each node contains feature information from multi-order neighboring nodes. Through the encoding layer, the node acquires its own weighted structural information. Through the decoding layer, the node obtains weighted structural information from nodes of different types. Ultimately, each node obtains an embedding with rich information. The entire process is illustrated in Eqs. (13), (14), and (15).

According to Eq. (13) we input each literal embedding into a three-layer perceptron $L_{vote}$ with a hidden layer size of 128 dimensions to extract the categorical information and obtain the vote value $L^{(T)}$ for each literal. Then, the positive and negative literals of the same variable are combined into a 2d-dimensional eigenvector of the variable $X^{(T)}$, and $X^{(T)}$ is averaged by rows, as in Eq. (14). Finally, the final prediction is obtained by the activation function sigmoid, as shown in Eq. (15). Our model trains the network by minimizing the cross-entropy loss between the predicted and true labels as in Eq. (16).

$$L^{(T)} = L_{vote}\left(L^{(t)'}\right). \tag{13}$$

$$x^{(t)'} = \mathrm{Mean}\left(\mathrm{concat}\left(L^{(T)}[0, n], L^{(T)}[n, 2n]\right)\right). \tag{14}$$

**Table 1  Setup of the dataset used for the experiment.**

| Datasets | Min_n | Max_n | train_n | val_n | eval_n |
|---|---|---|---|---|---|
| $SR_Q(3, 10)$ | 3 | 10 | 5000 | 3000 | 3000 |
| $SR_W(3, 10)$ | 3 | 10 | 300000 | 30000 | 30000 |
| $SR_W(10, 40)$ | 10 | 40 | 300000 | 30000 | 30000 |
| $SR_{cv}(3, 10)$ | 3 | 10 | 100 | 100 | 100 |
| $SR(20)$ | 20 | 20 | 100 | 100 | 100 |
| $SR(40)$ | 40 | 40 | 100 | 100 | 100 |
| $SR(60)$ | 60 | 60 | 100 | 100 | 100 |
| $3SAT_{100}$ | 100 | 100 | 8000 | 1000 | 1000 |
| $3SAT_{150}$ | 150 | 150 | 8000 | 1000 | 1000 |
| $3SAT_{200}$ | 200 | 200 | 8000 | 1000 | 1000 |
| $3SAT_{250}$ | 250 | 250 | 8000 | 1000 | 1000 |
| $3SAT_{300}$ | 300 | 300 | 8000 | 1000 | 1000 |

$$y = \text{sigmoid}\left(x^{(t)'}\right). \tag{15}$$

$$\text{LOSS} = -\frac{1}{N}\sum_{i=1}^{N}\left[y_i\log(p_i) + (1 - y_i)\log(1 - p_i)\right]. \tag{16}$$

$N$ is the number of SAT instances in the dataset. $y_i$ is the true label (0 or 1) for sample $i$. $p_i$ is the predicted probability (between 0 and 1) for sample $i$.

## EXPERIMENTAL SETUP

To ensure the reproducibility of the experiments, this section provides an explanation of the experimental setup conducted in this article. For each subsection, we will discuss the type of data used, the device setup, the list of parameters involved, the development of performance metrics, and the choice of baseline methodology.

### Simulation datasets

In this experiment, we generated two different SAT problems: the random SAT problem SR(n) and the random 3-SAT problem. To ensure fairness, all experimental datasets were generated at once, using the same dataset on both models without additional randomization, and each dataset had 50% of satisfiable and 50% of unsatisfiable instances. The specific setups for the 12 datasets are shown in Table 1.

### *Random SAT problem SR (n)*

Random SAT problems are often difficult to solve and are widely used to test the performance of SAT solvers and other optimization algorithms, and by comparing the performance of different algorithms on random SAT problems can better evaluate their performance in real problems.

$SR(n)$ represents a random SAT problem with $n$ variables. We adopt a NeuroSAT-like approach to generate data, which produces a pair of random k-SAT problems, one satisfiable and the other unsatisfiable. The specific generation procedure is as follows: (1) $SR(n)$ first samples a small integer $k$ (with a mean slightly above 4). (2) It uniformly randomly samples $k$ variables and negates each variable with a 50% independent probability. (3) It generates a random clause over $n$ variables and continues this process to generate the clauses $C_i$. (4) These clauses are added to the SAT problem instance and solved using the conventional solver MINISAT until the addition of the clause $C_m$ renders the problem unsatisfiable. $\{C_1, \ldots \ldots C_{m-1}\}$ is having a satisfiable assignment, and negating the individual literals in $C_m$ will also result in a satisfiable problem $\{C_1, \ldots \ldots C'_m\}$. (5) Eventually, SR(n) generates a pair of instances, satisfiable instance $\{C_1, \ldots \ldots C'_m\}$ and unsatisfiable instance $\{C_1, \ldots \ldots C_m\}$. Since the number of random clauses of length 2 affects the difficulty of SAT problems, to avoid this, we control the size of the random clauses by combining different distributions through Eq. (17). Bernoulli denotes bernoulli distribution and Geo denotes geometric distribution.

$$Y_x = 1 + \text{Bernoulli}(0.7) + \text{Geo}(0.4). \tag{17}$$

We generated small-scale random SAT problems SR(3,10) and SR(10, 40). SR(3,10) denotes that the smallest instance has three variables and the largest has 10 variables. SR(10, 40) denotes that the smallest instance has 10 variables and the largest has 40 variables. The properties of these problems are as follows: (1) The number of clauses per instance is variable, ranging from a few dozen to over 200. (2) Each clause contains a different number of literals, usually between two and seven. (3) The main difference between satisfiable and unsatisfiable instances is that only one literal is negated in the same clause. Due to the highly structured nature of SAT instances, changing one variable may lead to different results. In order to make the model more accurate in recognizing these structural features and to evaluate the model's ability to capture them, we conducted experiments using instances of thousands and hundreds of thousands, respectively.

We also generated some lightweight data to test the model's performance on problems of varying complexity. We usually use $CV$ to denote the complexity of a problem. $CV$ is the ratio of clauses to literals. The higher the value of $CV$, the more clause constraints there are in the formula and the more difficult it is to solve the instance. By default the generated datasets have a $CV$ greater than 5, in addition to this we generated datasets $SR_{cv}(3, 10)$, $SR(20)$, $SR(40)$, and $SR(60)$ with $CV$ equal to 3 and 4 to evaluate the performance of our model on SAT instances of varying complexity.

### Random 3-SAT problem

Random 3-SAT problems are a form of NP-complete problem, where each clause is randomly generated and consists of three variables or negations of variables. According to the theoretical analysis and experimental results, the random 3-SAT problem is almost certainly satisfiable when the $CV$ is greater than 4.267; it is almost certainly unsatisfiable when the $CV$ is less than 4.267. When the $CV$ is 4.267, the satisfiability of the random 3-SAT problem becomes very difficult to predict. So at the middle tipping point, we created

**Table 2 Description and setting of parameters.**

| Parameter | Parameter name | Parameter value/setting |
|---|---|---|
| $N$ | Number of SAT instances | / |
| $n$ | Number of variables in each instance | / |
| $m$ | Number of clauses in each instance | / |
| $CV$ | Ratio of clauses to number of variables | / |
| $T$ | Number of iterations for messaging | 26 |
| $lr$ | Learning rate | $10^{-5}$ |
| weight_decay | Weight decay parameter | $10^{-10}$ |
| num layers | Number of encoder layers | 4 |
| $h$ | Attention Heads | 8 |
| $d$ | The dimension of the vector | 128 |

**Notes.**
A slanted vertical bar (/) indicates different cases of values in the text.

5 datasets $3SAT_{100}$, $3SAT_{150}$, $3SAT_{200}$, $3SAT_{250}$, and $3SAT_{300}$, the number of variables are 100, 150, 200, 250, 300. The ratio of the number of clauses to the number of variables in each dataset is 4.26.

## Experimental setup

Our model is implemented using the PyTorch framework. All experiments were run on a device configured with an Intel(R) Xeon(R) Platinum 8260 CPU @ 2.30 GHz and an NVIDIA GeForce RTX 3090 GPU. In the data generation phase, GPUs cannot be used because the minisat solver uses a backtracking sequential algorithm, but during training and evaluation, our models are fully deployed on GPUs, making full use of GPU computational resources through parallelism.

## Hyperparametric configuration

For the message passing part of TG-SAT, we adopt the same configurations as NeuroSAT (*Selsam et al., 2018*): we set the embedding dimension of each node and hidden cell to for each MLP ($L_{msg}$, $C_{msg}$, $L_{vote}$) there are three hidden layers and one linear output layer. Regularization is performed using $l_2$. The weight decay parameter is $10^{-10}$. The number of iterations of message passing on each problem is set to 26. We refer to the parameter settings of *Shi et al. (2021a)* and set the number of encoder layers to two, four and eight, respectively, for the experiments, and we choose four as the number of encoder layers to synthesize the running time and accuracy. The number of attentional heads was set to eight using the work of *Vaswani et al. (2017)*. We also set it to four for comparison and found a decrease in the accuracy of the model. The dimensionality of the input features and output features of each layer of the attention network is 128. The model was trained using the ADAM optimizer with a learning rate of $1 \times 10^{-5}$, dropout of 0.6. The loss function uses the binary cross-entropy function. The specific settings are shown in Table 2. A slanted vertical bar (/) indicates different cases of values in the text.

Confusion Matrix

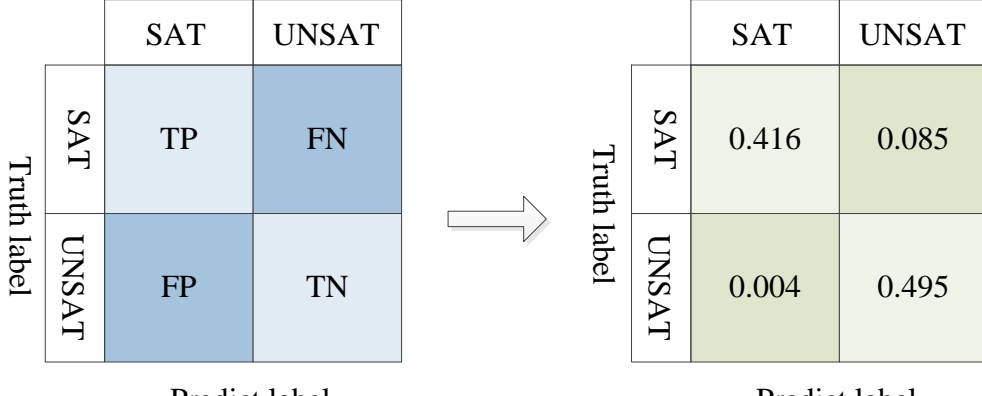

**Figure 6 Confusion matrix for binary classification.**

## Evaluation measures

Our model can be considered as a binary classification task. The effectiveness of the model will be evaluated by choosing the confusion matrix formula. According to *Sen & Deokar (2022)*, the confusion matrix for discrete classification is a two-by-two table formed by evaluating the number of four possible outcomes of the classifier. As shown in Fig. 6, the rows of the matrix represent the true values and the columns of the matrix represent the predicted values. $TP = 0.416$ signifies that 41.6% of satisfiable problems are correctly predicted, $FN = 0.085$ signifies that 8.5% of satisfiable problems are mistakenly predicted as unsatisfiable, $FP = 0.004$ signifies that 4% of unsatisfiable problems are erroneously predicted as satisfiable, and $TN = 0.495$ signifies that 49.5% of unsatisfiable problems are accurately identified.

We assessed the effectiveness of the models by calculating the accuracy(ACC) through the confusion matrix according to Eq. (18). In addition, we calculated precision(PRE), recall(REC), specificity(SPE), $F_1$-score($F_1$), and Matthew's correlation coefficient(MCC) to assess the performance of the neurosat and TG-SAT models in predicting problem satisfiability. These calculations are shown in Eqs. (19)–(23).

$$ACC = \frac{TP + TN}{TP + TN + FP + FN}. \tag{18}$$

$$PRE = \frac{TP}{TP + FP}. \tag{19}$$

$$REC = \frac{TP}{TP + FN}. \tag{20}$$

$$SPE = \frac{TN}{TN + FP}. \tag{21}$$

$$F_1 = \frac{2 \times \text{Precision} \times \text{Recall}}{\text{Precision} + \text{Recall}}. \tag{22}$$

$$MCC = \frac{TP \times TN - FP \times FN}{\sqrt{(TP+FP)(TP+FN)(TN+FP)(TN+FN)}}. \tag{23}$$

## Baseline methodology

Since the main focus of this article is on evaluating the performance of TG-SAT, we do not consider comparisons with traditional solvers or learning-based guided heuristic solver algorithms. These methods are not end-to-end solvers. We also do not consider other standalone SAT solvers (*Amizadeh, Matusevych & Weimer, 2018*; *Ozolins et al., 2021*; *Li et al., 2022*), because they do not produce a single binary result.

We compare TG-SAT with NeuroSAT, also an end-to-end neural network framework. Their input instances are both CNFs and both generate a single binary result. We performed experiments on random SAT instances of varying size and complexity, as well as on random 3-SAT problems of varying size. We employed 12 randomly generated datasets to evaluate the performance of our model. Detailed descriptions of these datasets are provided in Table 1 of 'Simulation datasets'.

NeuroSAT is a classical message-passing network that, while not as proficient as advanced solvers in solving SAT problems, can tackle larger and more challenging instances by simply executing more iterations beyond its training scope. It can also be extended to SAT problems with different structures, showcasing the potential of neural networks in addressing SAT problems. Our model builds upon NeuroSAT, aiming to enhance its capacity in capturing the rich generic and domain-specific structures inherent in SAT problems.

## RESULTS

### Experimental results

In this section, we discuss the model's prediction results for random SAT problems with different variable sizes and different complexities, as well as for random 3-SAT problems, respectively.

#### Results on SR(n) datasets with different variable sizes

On the $SR(3, 10)$ dataset, good accuracy can be achieved by executing only one epoch on both thousand and hundred thousand level data. We chose to analyze the results with 10 epochs executed. On the $SR_W(3, 10)$ dataset, NeuroSAT has a training accuracy of 0.854 and a testing accuracy of 0.816, whereas TG-SAT has a training accuracy of 0.950 and a testing accuracy of 0.957. On the $SR_W(10, 40)$ dataset, NeuroSAT has a training accuracy of 0.501 and a testing accuracy of 0.501, while the TG-SAT had a training accuracy of 0.854 and a testing accuracy of 0.867. The team working on NeuroSAT had trained on $SR(10, 40)$ with millions of instances and achieved an accuracy of 0.73–0.85. We trained TG-SAT with 300,000 instances for more than two weeks, and after 10 epochs, the test accuracy

**Table 3** Comparison of NeuroSAT and TG-SAT performance on $SR(n)$.

| Method | Dataset | Train | Val | Test |
|---|---|---|---|---|
| NeuroSAT | $SR_W(3,10)$ | 0.854 | 0.819 | 0.816 |
| | $SR_W(10,40)$ | 0.501 | 0.496 | 0.501 |
| | $SR_Q(3,10)$ | 0.935 | 0.933 | 0.931 |
| TG-SAT | $SR_W(3,10)$ | 0.950 | 0.957 | 0.957 |
| | $SR_W(10,40)$ | 0.854 | 0.867 | 0.867 |
| | $SR_Q(3,10)$ | 0.945 | 0.947 | 0.945 |

reached 0.867, while NeuroSAT's test accuracy on the same order of magnitude of data is only 0.501, which is clearly not as good as that of TG-SAT. this indicates that our model performs better in solving problems of much larger scale. The results are shown in Table 3.

To assess our model's performance stability, we conducted multiple training epochs on the simple and small dataset $SR_Q(3,10)$. Each training session involved 50 epochs, and we selected the best results for evaluation. The outcomes revealed that NeuroSAT achieved a training accuracy of 0.935 and a testing accuracy of 0.931, while TG-SAT attained a training accuracy of 0.945 and a testing accuracy of 0.945. Although TG-SAT only exhibited a marginal 0.014 improvement in accuracy over NeuroSAT on this basic dataset, our model demonstrated greater consistency and achieved higher accuracy in fewer epochs. As illustrated in Fig. 7, NeuroSAT's performance exhibited instability, requiring approximately 30 epochs to start converging for poor performance, and generally beginning to converge at 20 epochs for better performance. This suggests that NeuroSAT necessitates multiple message passing iterations to more precisely capture the structural features of SAT instances, thereby making its accuracy susceptible to the reliability of the information. Conversely, as depicted in Fig. 8, our model began to converge around 15 epochs and attained satisfactory accuracy more rapidly.

### Results on SR(N) datasets of varying complexity

We do experiments on four datasets $SR_{cv}(3,10)$, $SR(20)$, $SR(40)$, and $SR(60)$ to compare the prediction accuracies of NeuroSAT and TG-SAT for SAT instances with different levels of complexity. It can be seen through Figs. 9–11 that TG-SAT has higher accuracy than NeuroSAT on all datasets.

With Figs. 12 and 13 we can observe the effect of the size of the instance on the two models with the same level of complexity. For the simple instance of $CV = 3$, NeuroSAT produces similar results on datasets of different sizes, with an accuracy of about 90%. The accuracies of our models all exceed 90%. However, for complex instance with $CV > 5$, it can be clearly observed that the performance of NeuroSAT decreases as the instance size increases, while our model has more potential to solve problems of larger sizes.

With Figs. 14 and 15 we can observe the effect of the complexity of the instance on both models at the same scale. For instances of the same size, the performance of NeuroSAT decreases much more than that of TG-SAT as the $CV$ increases. This may be due to the fact that TG-SAT is able to utilize correlations between clauses and clauses, literals and literals, and clauses and literals. When the number of clauses and literals increases, our

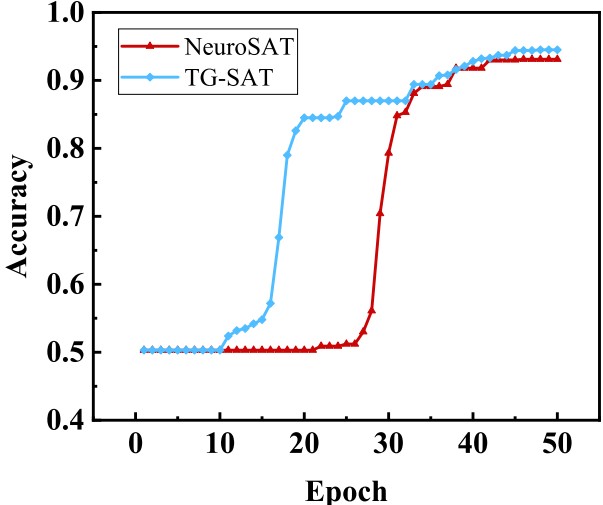

**Figure 7** Variation of accuracy with the number of epochs (poor).

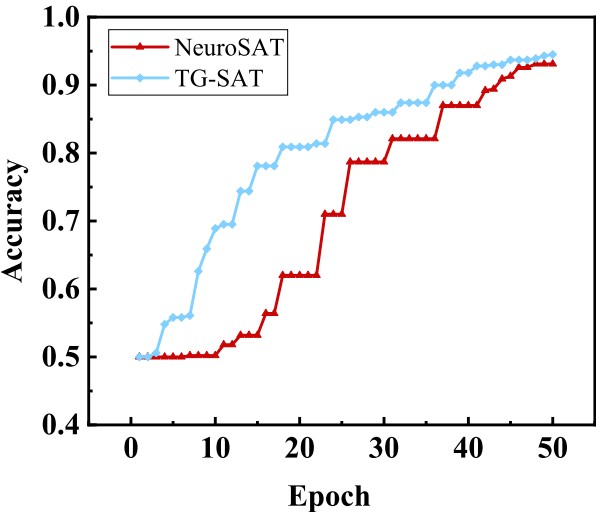

**Figure 8** Variation of accuracy with the number of epochs (good).

model can learn richer general and domain-specific structures and thus better utilize these correlations. In contrast, NeuroSAT learns instance-level features only through single-bit supervision and may not be able to obtain richer representations by reducing the instance size or decreasing the instance difficulty. Overall, our model TG-SAT outperforms NeuroSAT.

### Results on the random 3-SAT dataset

On the random 3-SAT problem, we trained and evaluated NeuroSAT and TG-SAT using five datasets $3SAT_{100}$, $3SAT_{150}$, $3SAT_{200}$, $3SAT_{250}$, and $3SAT_{300}$, respectively. The results

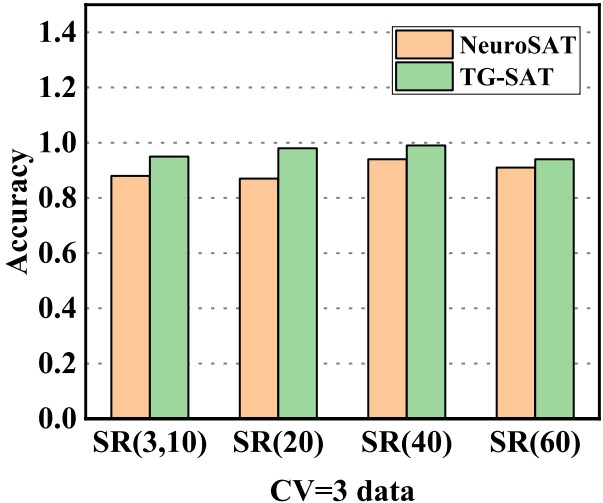

**Figure 9** Performance of the two models for different sizes of SAT problems with $CV = 3$.

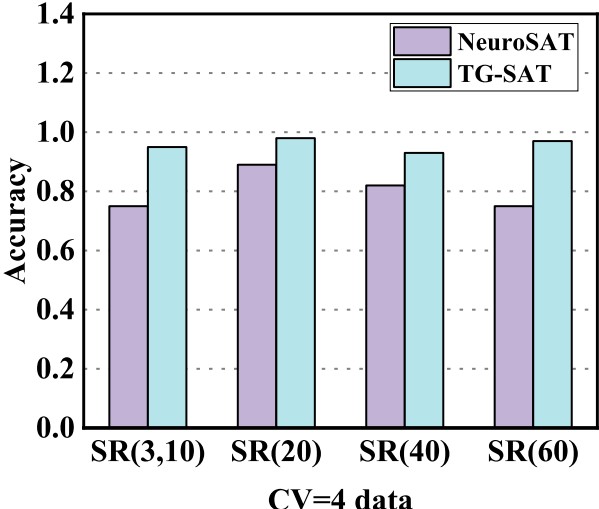

**Figure 10** Performance of the two models for different sizes of SAT problems with $CV = 4$.

are shown in Table 4. On the 100-variable dataset, NeuroSAT's test precision was 0.746 and TG-SAT's was 0.789; on the 200-variable dataset, NeuroSAT's test precision was 0.783 and TG-SAT's was 0.817; on the 300-variable dataset, NeuroSAT's test accuracy is 0.797 and TG-SAT's test accuracy is 0.828. The test accuracy of TG-SAT on all datasets improved by 2%–5% over the test accuracy of NeuroSAT.

To verify that TG-SAT can learn the general structural features of a given SAT problem instance, we test the model using instances of different sizes than those used in training to evaluate its generalization ability. We employ a model trained on SAT instances with 100 variables to predict the satisfiability of larger scale instances. Specifically, we test the

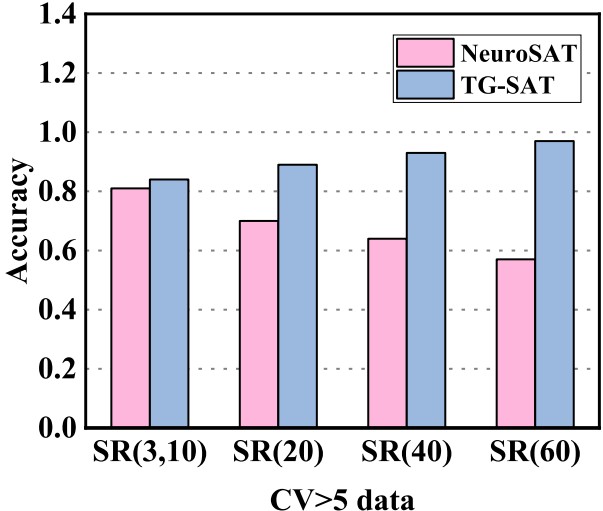

**Figure 11  Performance of the two models for different sizes of SAT problems with *CV* > 5.**

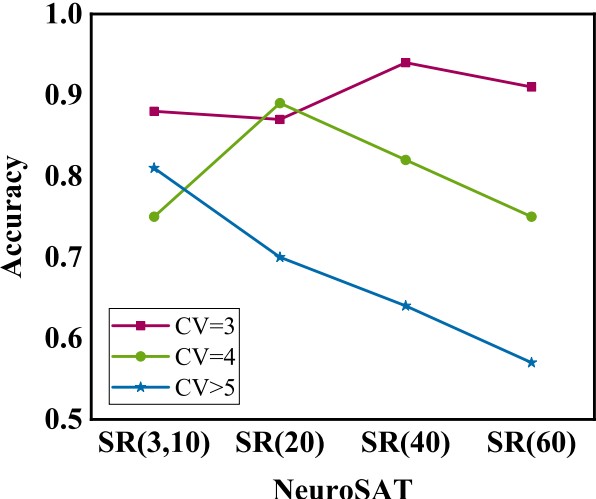

**Figure 12  Effect of the size of the SAT problem on the prediction accuracy of the NeuroSAT.**

model with instances having 150, 200, 250, and 300 variables, respectively. The results are summarized in Table 5. Even when trained solely on instances with 100 variables, both NeuroSAT and TG-SAT demonstrate good performance on other datasets. Although the accuracy of both models decreases compared to when trained and tested on instances of the same scale, TG-SAT achieves a test accuracy improvement of 1% to 6% over NeuroSAT when trained and tested on instances of different scales.

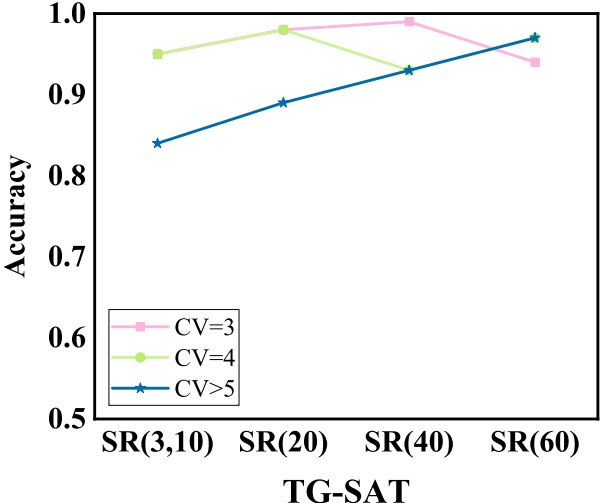

**Figure 13** Effect of the size of the SAT problem on the prediction accuracy of the TG-SAT.

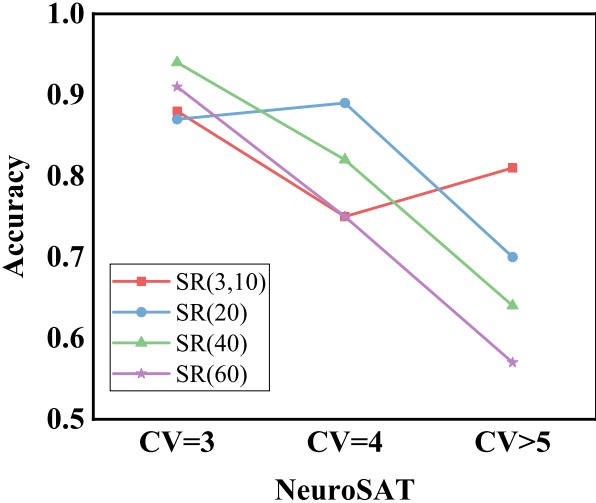

**Figure 14** Effect of SAT problem complexity on the prediction accuracy of the NeuroSAT.

## Performance evaluation

ACC is a metric that can assess the overall predictive accuracy of the model. However, for a more comprehensive evaluation of model performance, we consider multiple metrics. In addition to ACC, we also evaluate Precision (PRE), Specificity (SPE), Recall (REC), F1 score ($F_1$), Matthews correlation coefficient (MCC), and other metrics. These metrics provide insights into different aspects of model performance. We assess the experimental results on data from random SAT instances and random 3-SAT instances separately. The detailed results are presented in Table 6 for random SAT instances and Table 7 for random 3-SAT instances.

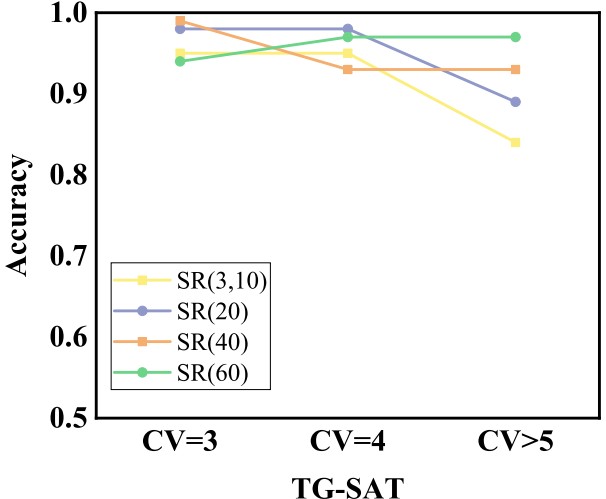

**Figure 15** Effect of SAT problem complexity on the prediction accuracy of the TG-SAT.

**Table 4** Experimental results on the random 3-SAT problem.

| Data | NeuroSAT | | | TG-SAT | | |
|---|---|---|---|---|---|---|
| | Train | Val | Test | Train | Val | Test |
| $3SAT_{100}$ | 0.762 | 0.750 | 0.746 | 0.879 | 0.786 | 0.789 |
| $3SAT_{150}$ | 0.769 | 0.766 | 0.764 | 0.826 | 0.782 | 0.800 |
| $3SAT_{200}$ | 0.784 | 0.774 | 0.783 | 0.823 | 0.805 | 0.817 |
| $3SAT_{250}$ | 0.812 | 0.801 | 0.788 | 0.835 | 0.822 | 0.814 |
| $3SAT_{300}$ | 0.805 | 0.820 | 0.797 | 0.834 | 0.839 | 0.828 |

**Table 5** Experimental results of training on 3-SAT problems with a variable size of 100, tested on other variable sizes.

| Vars | NeuroSAT-100 | TG-SAT-100 |
|---|---|---|
| 150 | 0.758 | 0.778 |
| 200 | 0.784 | 0.798 |
| 250 | 0.776 | 0.783 |
| 300 | 0.753 | 0.805 |

As can be seen from Tables 6 and 7, compared with NeuroSAT, the accuracy and F1 scores of our model are improved on all eight datasets, indicating that our model performs better in predicting both satisfiable and unsatisfiable instances, and the overall performance of TG-SAT is improved. TG-SAT has higher precision and specificity than NeuroSAT on five out of eight datasets, which also indicates that our model has better accuracy and reliability for categorizing positive and negative samples in general. The recall is higher than NeuroSAT on six datasets, indicating that our model is able to capture positive category samples effectively. The value of MCC ranges from −1 to 1, where 1 means perfect prediction, 0 means random prediction, and -1 means completely opposite prediction.

**Table 6   Comparison of ACC, PRE, and SPE on different datasets.**

| Data | ACC | | PRE | | SPE | |
|---|---|---|---|---|---|---|
| | NeuroSAT | TG-SAT | NeuroSAT | TG-SAT | NeuroSAT | TG-SAT |
| $SR_Q(3,10)$ | 0.931 | 0.945 | 0.998 | 1.000 | 0.998 | 1.000 |
| $SR_w(3,10)$ | 0.816 | 0.957 | 0.997 | 0.996 | 0.998 | 0.996 |
| $SR_w(10,40)$ | 0.501 | 0.867 | 0.497 | 0.946 | 0.624 | 0.956 |
| $3SAT_{100}$ | 0.746 | 0.789 | 0.774 | 0.808 | 0.674 | 0.722 |
| $3SAT_{150}$ | 0.764 | 0.800 | 0.845 | 0.829 | 0.849 | 0.806 |
| $3SAT_{200}$ | 0.783 | 0.817 | 0.887 | 0.853 | 0.915 | 0.865 |
| $3SAT_{250}$ | 0.788 | 0.814 | 0.740 | 0.857 | 0.681 | 0.863 |
| $3SAT_{300}$ | 0.797 | 0.828 | 0.739 | 0.788 | 0.643 | 0.737 |

**Table 7   Comparison of REC, $F_1$, and MCC on different datasets.**

| Data | REC | | $F_1$ | | MCC | |
|---|---|---|---|---|---|---|
| | NeuroSAT | TG-SAT | NeuroSAT | TG-SAT | NeuroSAT | TG-SAT |
| $SR_Q(3,10)$ | 0.865 | 0.889 | 0.927 | 0.941 | 0.871 | 0.895 |
| $SR_w(3,10)$ | 0.635 | 0.918 | 0.776 | 0.955 | 0.679 | 0.917 |
| $SR_w(10,40)$ | 0.375 | 0.764 | 0.427 | 0.845 | −0.002 | 0.734 |
| $3SAT_{100}$ | 0.798 | 0.837 | 0.785 | 0.822 | 0.475 | 0.563 |
| $3SAT_{150}$ | 0.692 | 0.795 | 0.761 | 0.812 | 0.543 | 0.599 |
| $3SAT_{200}$ | 0.653 | 0.770 | 0.753 | 0.810 | 0.588 | 0.637 |
| $3SAT_{250}$ | 0.893 | 0.768 | 0.809 | 0.810 | 0.588 | 0.633 |
| $3SAT_{300}$ | 0.940 | 0.913 | 0.828 | 0.846 | 0.615 | 0.663 |

In general, the closer the MCC value is to 1, the better the prediction performance of the model. Our model has MCC values greater than 0.5 on all 8 datasets, especially on small instances of the SAT problem. Overall, our model outperforms NeuroSAT on several evaluation metrics.

## Effectiveness of transformer and GRU

The traditional Transformer approach solves the SAT problem by computing weighted feature vectors for each node through self-attention scores. The self-attention layer utilizes relationships between nodes to determine which nodes are more important, thereby better capturing contextual information and dependencies. After multiple self-attention layers, the encoder converts the input sequence into a set of hidden states, where each hidden state contains relevant information about the input sequence. However, a limitation of this method is that the obtained node feature representations lack information about multi-order neighboring nodes. To address this limitation, we incorporate a message passing step after computing self-attention. This ensures that the resulting node feature representations contain information about their multi-order neighboring nodes.

We know that for any SAT instance, several key variables can be obtained through Boolean constraint propagation, and the assignment of other variables does not affect the satisfiability of the problem. In the message passing process, we aggregate information from

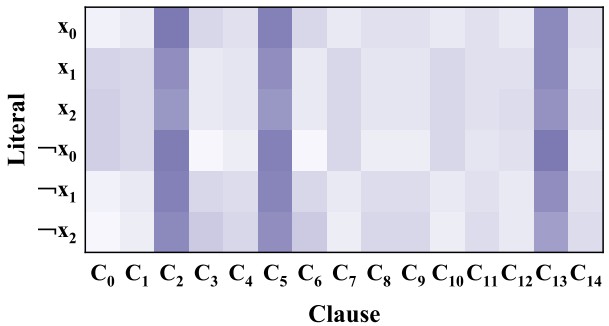

**Figure 16** Visualization of attention scores when calculating self-attention for a small problem.

neighboring nodes using graph neural networks. However, the node vector representations obtained in this way cannot leverage information from past nodes. To address this issue, we use GRU to update the feature representations of nodes. This ensures that the obtained node feature representations not only contain contextual information but also incorporate information about long-term dependencies between nodes. We prioritize assigning values to key variables to expedite the model's predictions. To identify these influential variables, after the message passing phase, we compute cross-attention scores between literal nodes and clause nodes at the decoding layer. Each node is assigned different weights based on these scores, guiding the nodes' voting process. If these few variables that significantly influence the satisfiability of the SAT problem are correctly assigned values, then the SAT problem is satisfiable, and a solution can be found.

To illustrate the role of the Transformer in updating node embeddings more clearly, we visualize the attention scores of adjacent literal nodes and clause nodes before and after message passing in an SAT instance with three variables and 15 clauses. The visualization of the self-attention scores before message passing is depicted in Fig. 16. In this visualization, the horizontal coordinates represent clause nodes, while the vertical coordinates represent positive literals in the first three rows and negative literals in the last three rows. The color of each intersection indicates the importance of the literal in adjacent clauses, with darker colors indicating higher importance. We observe that three clauses ($C_2$), $C_5$, and ($C_{13}$) correspond to all literals that are significantly darker in color. We speculate that after computing self-attention, the literal nodes correlate with the positions of all nodes to capture the contextual information of the entire instance. Key literals can be identified by detecting clauses that are likely to conflict based on the importance of most literals in those clauses. The visualization of the cross-attention scores after message passing is presented in Fig. 17. We argue that TG-SAT communicates not only within homogeneous nodes but also among heterogeneous nodes (between literals and clauses) through cross-attention. This computation of attention scores among neighboring nodes assigns higher weights to key literal nodes and provides rich semantic information for nodes to update their states. Consequently, the satisfiability of SAT problems can be predicted faster and more accurately.

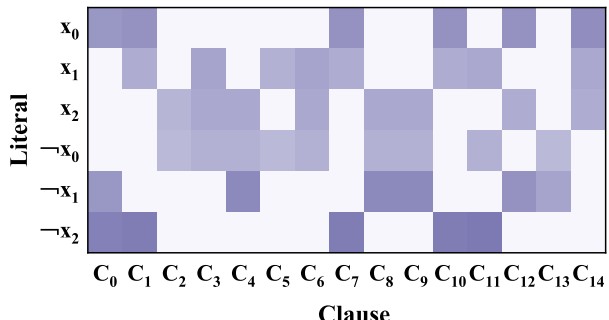

**Figure 17** Visualization of attention scores when calculating cross-attention for a small problem.

**Table 8  Results of ablation experiments.**

| Variants | $SR_Q(3, 10)$ | $3SAT_{100}$ |
|---|---|---|
| Transformer | 0.500 | 0.583 |
| MPNN | 0.739 | 0.745 |
| MPNN+GRU | 0.829 | 0.758 |
| MPNN+GRU+Cross-Attention | 0.902 | 0.775 |
| TG-SAT | 0.937 | 0.789 |

## Ablation study

In the previous section, we verified the effectiveness of our proposed model. In our model, Transformer and GRU are used for feature extraction. In order to validate the effectiveness of our model to extract features, we set up five sets of experiments, changing only the part of the model that extracts features. The first group is without message passing and only uses Transformer to compute self-attention to update the node embedding. The second group is using only the message-passing network. The third group adds GRU to the message-passing network. The fourth group was to add cross-attention to the third group. The fifth group is our proposed TG-SAT. We train on two datasets, $SR_Q(3, 10)$ and $3SAT_{100}$, respectively, and the experimental results are test results as shown in Table 8.

As we speculated, Transformer can capture contextual information, but it is difficult to capture information about multi-order neighbors, and it is also clear from the results that it is not a suitable approach to solve the SAT problem. The message passing model can aggregate and update the information of multi-order neighbor nodes to learn the rich semantic information of the SAT problem, but it cannot judge whether the acquired information is useful or not. The addition of GRU effectively solves this problem by selectively capturing important information. Cross-attention is used to update the weighted embeddings of different attribute nodes, and our model TG-SAT obtains the highest accuracy. Experiments show that our improved model using Transformer and GRU plays an important role in improving the accuracy of prediction for SAT problems, especially for difficult SAT problems.

## CONCLUSION

In recent years, Transformer has demonstrated remarkable performance in various tasks within natural language processing and computer vision, while GRU has proven effective in processing time series data with long-term dependencies. In this article, we introduce TG-SAT, a novel framework for predicting the satisfiability of random SAT problems, by integrating GRU modules into the Transformer architecture for the first time.Firstly, TG-SAT leverages the self-attention mechanism to obtain global and contextual information of all nodes, enabling preliminary predictions regarding which clauses are more likely to be in conflict. Secondly, multi-order neighbor information and historical data are updated through a message passing network and GRU. Finally, the cross-attention mechanism is employed to focus on critical literal nodes. In predicting the satisfiability of SAT problems, our study emphasizes the feature representation of important literal nodes. Our findings demonstrate that our model effectively learns and captures invariant structural features of key nodes, leading to improved representation of the feature space of Boolean formulas, particularly beneficial for handling complex Boolean formulas. Notably, our model consistently outperforms NeuroSAT in terms of prediction accuracy on both random SAT problem and random 3-SAT problem datasets. Moreover, our model learns the representation of the SAT problem and model parameters directly from raw data, eliminating the need for manual feature design or rule creation. This end-to-end approach enhances adaptability to SAT problems of varying sizes and complexities and exhibits superior generalization capabilities.

This article represents the first endeavor to apply the TG-SAT model for predicting the satisfiability of Boolean formulas. Further experimental validation is warranted to explore the impact of different parameter combinations on the model. Future work will delve into investigating the process of feature selection and parameter optimization. Additionally, for instances predicted to be satisfiable, we will explore machine learning methods for solving them.

### Funding

This work has been supported by the National Natural Science Foundation of Chinaunder Grant No. 62202145 and No. 61972134, the Young Elite Teachers in Henan Province No. 2020GGJS050, the Doctor Foundation of Henan Polytechnic University under Grant No. B2020-31, and the Innovative and Scientifc Research Team of Henan Polvtechnic University under No. T2021-3. The funders had no role in study design, data collection and analysis, decision to publish, or preparation of the manuscript.

### Grant Disclosures

The following grant information was disclosed by the authors:
The National Natural Science Foundation of Chinaunder: No. 62202145, No. 61972134.
Young Elite Teachers in Henan Province: No. 2020GGJS050.

Doctor Foundation of Henan Polytechnic University: B2020-31.
Innovative and Scientifc Research Team of Henan Polvtechnic University: No. T2021-3.

## Competing Interests

The authors declare there are no competing interests.

## Author Contributions

- Wenjing Chang conceived and designed the experiments, authored or reviewed drafts of the article, and approved the final draft.
- Mengyu Guo conceived and designed the experiments, performed the experiments, analyzed the data, performed the computation work, prepared figures and/or tables, and approved the final draft.
- Junwei Luo conceived and designed the experiments, authored or reviewed drafts of the article, and approved the final draft.

## Data Availability

The data is available at figshare: Guo, Mengyu (2023). 数据.rar. figshare. Dataset. https://doi.org/10.6084/m9.figshare.24658437.v1.

## Supplemental Information

Supplemental information for this article can be found online at http://dx.doi.org/10.7717/peerj-cs.2169#supplemental-information.

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
