# Peer review of "Predicting the satisfiability of Boolean formulas by incorporating gated recurrent unit (GRU) in the Transformer framework"

_PeerJ Computer Science, doi:10.7717/peerj-cs.2169_

## Round 0.1 · original submission · Major Revisions

Dear Authors, according to the comments provided by the reviewers, please follow the suggestions to revise and improve the overall quality of the paper.
In what follows some relevant comments extracted from those provided by reviewers:

- Grammar and Readability: The paper requires a revision of the English, and some parts are unclear.

- Clearly revise the description and the references on the SAT problem

- The innovation of the paper seems limited. The proposed method is a straightforward combination of existing techniques, which makes it less innovative. Also, more details about the proposed method should be provided.

- It is necessary to better motivate the study and the proposal

- Reviewer 3 highlights several critical points in the presentation of the proposal, most on the first pages

- The evaluation is weak. Please consider using a more convincing way to evaluate the proposed method, e.g., comparing the approach with SOTA models

- Reviewer 2 highlights that the experimental setup of the conducted simulation is too simple and vague, and suggests the author include a section on Experimental Setup which includes the following: (a) Simulation Design (b) Dataset (c) Parameter Assignments (d) Formulations of the Performance Metrics (e) Reproducibility of the proposed model (f) Baseline methods

- There are several papers that have addressed similar problems, but it is necessary to further highlight the novelty between the proposed study and the related literature.

- I suggest adding a clear research objective or research questions in the introduction section and specifying what the main research problem or hypothesis is addressed

- The discussion of the results does not highlight the strengths and weaknesses of the proposed approach.

Concluding Remarks:

I think that the paper could be improved with the considerations that reviewers reported in the review, but this version is not ready for publication

**Language Note:** The Academic Editor has identified that the English language must be improved. PeerJ can provide language editing services - please contact us at [email protected] for pricing (be sure to provide your manuscript number and title). Alternatively, you should make your own arrangements to improve the language quality and provide details in your response letter. – PeerJ Staff

Reviewer 1 ·

Basic reporting

Line 142 exceeds the page margins

Experimental design

no comment

Validity of the findings

no comment

Additional comments

The various sections of the paper are clearly written. The strategy seems to be valid and interesting. The results and comparison with the NeuroSAT algorithm are clearly described.

Reviewer 2 ·

Basic reporting

This paper proposed Predicting the satisfiability of Boolean formulas by incorporating GRU in the transformer framework. This paper require major revisions.

1. The Introduction is very vague. It is full of citation without proper justification. The author should discuss in detail each citation that contributed to the work.

2. The author should include more recent citations in the Introduction section.

3. I strongly suggests the author to add the following works in the Introduction/Related Work section as these works also discussed the study of ANN-SAT.
(a) Weighted random k satisfiability for k= 1, 2 (r2SAT) in discrete Hopfield neural network
(b) Random satisfiability: A higher-order logical approach in discrete Hopfield Neural Network
(c) Random Maximum 2 Satisfiability Logic in Discrete Hopfield Neural Network Incorporating Improved Election Algorithm

4. The author may add the organization of the paper at the end of the Introduction section.

Experimental design

1. I suggest the author include a schematic diagram of the proposed model. The author may refer/cite to the following work regarding this matter.
(a) YRAN2SAT: A novel flexible random satisfiability logical rule in discrete hopfield neural network

2. The experimental setup of the conducted simulation is too simple and vague. I suggest the author include a section on Experimental Setup which include the following:
(a) Simulation Design
(b) Dataset
(c) Parameter Assignments
(d) Formulations of the Performance Metrics
(e) Reproducibility of the proposed model
(f) Baseline methods

3. Can the author justify how the parameters in the experiment was set/chosen?

Validity of the findings

1. The author should include more evaluation measures such as Precision and Sensitivity.

2. The author should compare the results with other existing works. The author may refer to the works mentioned in 3(a) & 3(b).

3. The author should consider Friedman test to validate the superiority of the proposed model. Kindly refer/cite the following paper on how to do the Friedman test
(a) A modified reverse-based analysis logic mining model with Weighted Random 2 Satisfiability logic in Discrete Hopfield Neural Network and multi-objective training of Modified Niched Genetic Algorithm

Additional comments

1. Any reason why MLP was chosen and not any other ANN?

2. CNF SAT with third order logic is classified as an NP problem. Why the author did not consider P classified SAT structures such as 2SAT or Random 2 Satisfiability (RAN2SAT)?

Reviewer 3 ·

Basic reporting

From my point of view, the quality of writing is not good enough for publishing the paper as is. I understand, that the authors are not native English speakers, but still, there are places in the paper which are barely comprehensible due to the awkward phrasing.
For example, the authors often use the phrase "SAT problem", referring to SAT as a problem, to SAT instances, and probably to something in the middle.
The Boolean satisfiability problem (SAT) is formulated as "For an arbitrary Boolean formula to decide whether it is satisfiable". An instance of SAT: a formula, in CNF or any other form - is a SAT instance. It is not proper to refer to it as a "SAT problem".
In the abstract, the authors say that "traditional algorithms can only solve specific types of [SAT] problems". This is a strong statement, which requires justification (if the authors are able to provide it).
The next sentence says that "The ability of neural networks to ... is key to solving SAT problems". Really. Do the SAT solver developers know? No solver in SAT competition uses NN in any form. Not one.

In lines 25-26 "Any problem that requires decision making under a set of conditions can be constructed as a SAT problem". What in the world should that mean? The statement is a bit too ambiguous to decipher.

The "Non-deterministic Polynomial-complete (NP-C)" - I see such a wording for the first time in my life, and I have seen my share of papers. The citation in line 31 on the seminal paper by Cook is improper., It does nothing towards finding approximation and heuristic algorithms for SAT. This is a classical work that showed that SAT is NP-complete.

lines 38-40 "This type of approach to improve the incomplete algorithm, although a little more efficient in solving, cannot be proved unsatisfiable for some instances": the authors mean to say that incomplete algorithms for SAT can not prove unsatisfiability. Can anyone percieve this from the cited sentence? I have my doubts.

" Additionally, there are methods for enhancing the completeness algorithm." An algorithm is either complete or incomplete. It is impossible to make a complete algorithm more complete, or an incomplete algorithm more incomplete.

This is just the first page! Do I even need to go further?

On second page, in Related work, lined 86 -95 on DP, DPLL and CDCL. The authors say, that DPLL algorithm inroduces non-deterministic search. Have they even looked at how DPLL works? If yes, then please look again. Learning in DPLL? no way.
"learning clause" is "clause learning" as in "Conflict-Driven Clause Learning" which is written in the same line.
"Improves the accuracy of reasoning and searching". What accuracy are we talking about? It is a complete algorithm for a discrete problem.
Citation of MiniSat paper and LCM paper is weird. Why not cite the handbook of satisfiability for gods sake?
line 102, Hordesat does not use machine learning as far as I remember. It is a pretty standard portfolio solver.
line 104 "SAT problem determination" - I can't even guess what it is.

I will not proceed to other papers, I have at least 5 marks on each page without even reading with any degree of attentiveness.

The part on the description of the system looks a bit better, but still suffers from poor language.

Overall, the idea of the paper is more or less okay, the authors propose to extend the GNN-based SAT solver by the GRU block and show that it brings improvements compared to NeuroSAT on random benchmarks.
However, the text is in an awful condition, I can not give it a positive evaluation.

Experimental design

The experiments look more or less okay. I dislike that the authors only consider the random benchmarks, and do not compare with traditional solvers or at least other nonstandard solvers, like MatSAT or some other GNN-based solvers (which all stem from NeuroSAT, but still differ from it).
I could not comprehend the part in lines 358-359 on "the few variables that affect the satisfiability of a SAT problem are satisfiable". I have no idea what it means.

The ablation study typically means removing a single part from the whole to investigate how it affects the performance, but the authors do it the other way. It is acceptable, I think, but the standard ablation study would be ok as well.

Validity of the findings

I think that the findings in the paper are most likely valid. The problem is that one needs extraordinary patience and a lot of time to comprehend what authors wanted to convey by their text, since it is of a very poor quality.

Additional comments

My recommendation is to revise the paper. Ideally, the authors should either make an effort to read other papers on the topic and imitate their language, or to find a colleague that can consult them how to make the manuscript more comprehensible. The paid services are unlikely to be of any use since their translators tend to not have the specific scientific knowledge required to narrate the study properly.

---

## Round 0.2 · Minor Revisions

Reviewer 3 has provided different issues to address for improving the overall quality of the paper regarding language issues and structure, figures, and tables. It is necessary to follow the suggestion of the reviewer in this round and in the previous one.

Reviewer 1 ·

Basic reporting

No comment

Experimental design

No comment

Validity of the findings

No comment

Additional comments

My opinion is that the authors have succeeded in making the improvements suggested by the reviewers, thus increasing the quality of the work. The work complied with the requirements, being interesting and well described in my opinion.

Reviewer 2 ·

Basic reporting

The author has complied with all my reviews.

Experimental design

The author has complied with all my reviews.

Validity of the findings

The author has complied with all my reviews.

Additional comments

The author has complied with all my reviews.

Reviewer 3 ·

Basic reporting

I review the paper for the second time. During the first pass, my main concern was the quality of the text. During the revision process, the authors improved on it, somewhat. Despite that, there are still a lot of language issues, but probably not as pronounced as in the previous iteration.

I find the references more or less satisfactory, background and context are presented in favour of the end-to-end NN-based SAT solving, but that is to be expected, since NN SAT solvers can not compete with CDCL on the benchmarks, where CDCL shows good results.

I also have no qualms with regards to paper structure, figures and tables.
The necessary terms are introduced in time, everything is more or less ok.

Some of the minor issues:
line 11: ... can be used as more general algorithms - compared to what?
lines 20-21: SAT questions - there is no such term in Boolean satisfiability.
line 21: accuracy of random 3-SAT -> accuracy on random 3-SAT
line 23: outperforms other neural network methods - I fail to see the comparison to anything besides NeuroSAT.

line 25: in SAT problems, -> In SAT
line 26-27:the goal of the problem is to find the assignment of a set of variables such that the expression is true. - or to prove that it does not exist. This is an important part. A decision problem implies a "Yes or No" answer. Not "Yes or UNKNOWN".
line 32 This traditional solver - what traditional solver do authors allude to?
lines 33-34 SAT problem instances -> SAT instances.

lines 49-50 These methods play a crucial role in creating interpretable artificial intelligence. - What methods? In the previous sentences the authors speak about incomplete SAT solving methods. They have no place in XAI because XAI requires the ability to prove unsatisfiability as well, at least in some of the tasks.

line 58 satisfiable features - I work in this area for a long time and don't know what features are these.

lines 108-109 in predicting SAT problems - What does that mean?

line 161 machine automatically generated stochastic -> randomly generated

line 186 \{x_i\}_{j=1}^J - what is the index "i" for then?
line 187 - the set of logical operators can consist of any operators. The basis does not even have to be complete, in general. It is not limited to conjunction disjunction and negation either. No citation for the paper on Tseitin transformations.

line 193: To want -P to be true > In order for P to be true

Frankly, I got tired from listing even these issues, I did not want to go further.

Experimental design

The authors opt to compare to a single solver, neuroSAT, with unclear motivation expressed in lines 389-390. I can not say that I understand the idea behind this restriction, but whatever. I do not really expect miracles here.
Yet, I fail to see why the authors do not compare with other neural SAT solvers. They write that, say the solver from Ozolins et al. is not an end-to-end neural SAT solver, so what? I think that the authors could have combined their solver with e.g. MiniSat and compared with more competitors.

I do not see it as a deficiency only because I do not expect any miracles from end-to-end neural SAT solvers. They managed to improve on neuroSAT, okay, that is an acceptable result.

The investigation is performed quite well, includes the ablation study, I do not have any major issues with the experimental design overall.

Validity of the findings

I believe that the findings are mostly valid. I do not think that they are very impactful, but they still contribute to the area of knowledge.

Additional comments

I appreciate that the authors revised a lot of text.
However, it is still not sufficient. Overall, the findings are ok. The experiments are obviously cherry-picked but still sufficient, taking into account the peculiarities of NN-based SAT solvers.
But the presentation is not okay. The quality of the language is bad. It needs to be improved. At least to a satisfactory level. At least to the level when a reviewer can read a couple of pages and not find any issues.

I urge the authors to again proofread the text and ideally ask for professional help with fixing straightforward language-related issues.

---

## Round 0.3 · accepted · Accept

The paper has been sufficiently improved and it's ready for the publication